# The Optimized Design of Soil-Touching Parts of a Greenhouse Humanoid Weeding Shovel Based on Strain Sensing and DEM-ADAMS Coupling Simulation

**DOI:** 10.3390/s24030868

**Published:** 2024-01-29

**Authors:** Jianmin Gao, Zhipeng Jin, Anjun Ai

**Affiliations:** School of Agricultural Engineering, Jiangsu University, Zhenjiang 212013, China; 2212116042@stmail.ujs.edu.cn (Z.J.); 2221916001@stmail.ujs.edu.cn (A.A.)

**Keywords:** humanoid, correspond motion trajectory, the conditions of entry and cutting, simulation system, strain measurement

## Abstract

To overcome the shortcomings of plowing and rotary tillage, a human-like weeding shoveling machine was designed. The machine’s various moving rods were analyzed using Matlab R2019b(9.7.0.1190202) software to determine the appropriate entry and cutting conditions, as well as non-cutting conditions. It was concluded that a θ2 of 90° was optimal for cutting the soil and that the shoveling depth was suitable for greenhouse weeding. The Adams and DEM coupled discrete element simulation system was developed for this machine and was used to analyze the rotating shaft torque and shovel bending moment. A strain measurement system based on strain gauges was designed to measure the rotating shaft torque and shovel bar bending moment. A bending moment and torque measurement system was designed to perform field measurement tests for comparison with simulation results. The simulation system’s rotating shaft had an average torque error of 6.26%, while the shovel rod’s bending moment had an average error of 5.43%. The simulation accuracy was within the acceptable error range. Table U8 (81 × 44) of the Uniform Design of the Mixing Factor Level for the Homogeneous Virtual Simulation Test includes eight levels of forward machine speed ranging from 0.1 to 0.45 m/s and four levels of output shaft speed ranging from 90 to 165 r/min. Crank lengths were set at four levels ranging from 155 to 185 mm, while shovel lengths were set at four levels ranging from 185 to 230 mm. Four types of shovel shapes were proposed, including pointed curved shovels, pointed straight shovels, straight-edged curved shovels, and straight-edged straight shovels. A mathematical model was created via a regression analysis of the results of coupled simulation tests to establish the relationship between shaft torque and shovel rod bending moment, tool advance speed, shaft speed, crank length, tool length, and tool shape. The model was used to determine the optimum working parameters.

## 1. Introduction

Soil tillage operations mainly use rotary tillers and plowing operations, both of which form a solid layer at the bottom of the tillage layer [1,2,3], and this plow sole hinders the normal development and growth of crop roots, making it impossible for them to penetrate ward to absorb water and nutrients, resulting in a decrease in crop yield [4]. At the same time, the plow sole will slow down the infiltration of rainwater, and it is easy for surface runoff to occur during heavy rainfall. This results in surface nutrient loss, soil water erosion, and reduced soil fertility, and farmland quality is thereby reduced [5,6,7,8]. Grass easily wraps around the knife during the rotary tillage operation, and a serious clay phenomenon results from the plowing operation.

In view of the current insurmountable problems of grass entanglement and clay in tillage machinery, scholars at home and abroad have performed relevant research, such as Salokhe V M’s comparative test of reverse spoon-type new rotary tillage knives and traditional C-type rotary tillage knives in viscous wetlands and dryland tests. The results show that there is greater use of reverse spoon-type new rotary tillage knives in wetlands than that of traditional C-type rotary tillage knives in clay, while the opposite trend prevails in drylands [9,10]. SağLam M et al. analyzed the effects of different tillage methods, such as plowing and rotary tillers and adhesion of soil to equipment parts on contact, and the results showed that for operations in heavy cohesive soils, a reduced rotary tillage operation should be adopted [11]. Foreign scholars have been inspired by animal claws and feet; the use of bionic electron infiltration technology to design farming tools to reduce soil adhesion has attracted the attention of scholars.

In overcoming the problem of tillage machinery and weeding, scholars have proposed partial solutions. In 1998, Wang et al. [12], in Japan, investigated a way of preventing the machine from adhering to the soil by using sinusoidal vibration and carried out a comparative test using different soils; the test results show that the adhesion rate of soil and metal contact plate is closely related to the vibration acceleration; when the vibration frequency is in the range of 60~100 Hz the soil detachment rate reaches up to 100%, and when the vibration frequency is less than 50 Hz or greater than 100 Hz, the soil adhesion rate is obviously reduced, but have greater limitations. Yoshiyuki O proposed a new rotary tillage device and operation method to effectively prevent field weeds and straw from becoming entangled on rotary tillage blades [13]; however, the device is cumbersome and consumes a lot of power. Masakuni I et al. proposed a rotary tiller plow shaft anti-winding device to solve the problems of weed winding between the end of the tillage shaft and the side of the bearing (transmission, etc.) and the winding of the rotary tillage tool [14]. Wakuta T proposed an anti-entanglement guide at the end of the shaft, designed to fix the detachable protective device to both ends of the shaft of rotation, and each shield prevents weeds and straw from winding around the shaft end [15], but high power consumption. Zhuang Yueqin et al. applied a rotary cultivator with a staggered structure of positive and negative rotation of double-shaft cutterheads, and the test results showed that the machine achieved the purpose of sufficient shearing of soil, weeds and root stubble, which better prevented the occurrence of knife roller clay adhesion and grass entanglement [16]. Gao Lihong proposed avoiding entanglement by reasonably arranging the arrangement of the row of rotary tillage knives under the premise of ensuring the tillage conditions [17]. Zheng Kan et al. proposed a resonant source device to form a rigid connection with the anti-entanglement sticky knife group, and the vibration of the resonance source device was transmitted to the anti-entanglement sticky knife group to reduce the sticking phenomenon of soil and weeds [18]. With the development of new material technologies widely used in soil work components, Soni et al. [19], in 2007, applied high molecular weight polyethylene (UHMW-PE) to the surface of a plow blade based on its hydrophobicity. In order to study the effect of the shape and size of the coating on the operational resistance of the plow blade and soil adhesion, five different shapes of the cover layer were designed to establish a proportional relationship between the height of the dome and the diameter of the base to characterize the adhesion of the soil, and the test results showed that when the ratio of the height of the dome to the diameter of the base was ≤0.5, the operational resistance could be reduced by 10–30%, and the soil adhesion normal to the soil could be reduced by 10–60%. In 2016, Barzegar M et al. [20] utilized the low friction and self-scouring properties of the high-molecular-weight polyethylene (UHMW-PE) material as a coating on the surface of tillage knives in order to reduce the adhesion of the soil during the operation. The test results showed that UHMW-PE-coated steel can reduce traction force and lower power consumption compared to mild steel work of the same size and shape, thus improving tillage efficiency. Massah J et al. [21] used the property of non-adhesion of the body surfaces of animals living in the soil to derive a bionic electro-osmosis technology that can be used to reduce the rate of adhesion between tillage machinery and the soil. By designing and developing electro-osmotic plates with different geometries, the effects of the area ratio of electrodes, voltage, and current application time on the adhesion of will to the soil were investigated. The test results show that the use of bionic electro-osmosis technology effectively reduces the adhesion rate by more than 29.8 percent compared to other traditional metal plates, which can be used on tillage equipment to achieve the purpose of reducing adhesion. However, the use of new materials is costly, environmentally impactful, and technically demanding, which discourages farmers from using them.

According to the comprehensive analysis of the existing viscosity reduction and entanglement avoidance technology, many mechanisms are still under exploration, and the problem of clay adhesion to the working tool cannot be better solved, and the phenomenon of grass entanglement still occurs in the actual operation process. Based on the above analysis, this paper reports on the study of a human-like shoveling machine to overcome the limitations of current rotary tilling machines. To overcome the shortcomings of plowing and rotary tillage, a human-like weeding shoveling machine was designed. The machine’s various moving rods were analyzed using Matlab R2019b(9.7.0.1190202) software to determine the appropriate entry and cutting conditions, as well as non-cutting conditions. The Adams and DEM coupled discrete element simulation system was developed for this machine and was used to analyze the rotating shaft torque and shovel bending moment. A strain measurement system based on strain gauges was designed to measure the rotating shaft torque and shovel bar bending moment. A bending moment and torque measurement system was designed to perform field measurement tests for comparison with simulation results. Table U8 (81 × 44) of the Uniform Design of the Mixing Factor Level for the Homogeneous Virtual Simulation Test includes eight levels of forward machine speed ranging from 0.1 to 0.45 m/s and four levels of output shaft speed ranging from 90 to 165 r/min. Crank lengths were set at four levels ranging from 155 to 185 mm, while shovel lengths were set at four levels ranging from 185 to 230 mm. Four types of shovel shapes were proposed, including pointed curved shovels, pointed straight shovels, straight-edged curved shovels, and straight-edged straight shovels. A mathematical model was created via a regression analysis of the results of coupled simulation tests to establish the relationship between shaft torque and shovel rod bending moment, tool advance speed, shaft speed, crank length, tool length, and tool shape. The model was used to determine the optimum working parameters.

## 2. The Working Principle and Structure of the Human-like Shoveling Machine

Manual shoveling is mainly divided into the stages of soil entry, shoveling, throwing, and resetting, with this operating process being periodic. The operation process is shown in Figure 1. The process of the shovel knife entering the soil process is shown in Figure 1a. Figure 1b shows the shoveling process, and Figure 1c shows the throwing process. Figure 1d shows the blade shoveling reset process. Manual shoveling is mainly divided into the stages of soil entry, shoveling, throwing, and resetting, with this operating process being periodic. The operation process is shown in Figure 1. The process of the shovel knife entering the soil process is shown in Figure 1a. Figure 1b shows the shoveling process, and Figure 1c shows the throwing process. Figure 1d shows the blade shoveling reset process.

Manual shoveling of soil is approximately regarded as working uniformly along a straight line, and the knife is shoveled according to the regular movement stroke. Then, according to the operation process, the movement trajectory of the shovel knife can be obtained, as shown in Figure 2, where V represents the speed of human movement, S represents the spacing of the shovel, and the arrow direction indicates the direction of the shovel knife movement.

According to the manual shoveling operation process, the positive view of the motion trajectory of a single operation cycle of the shovel is shown in Figure 3 to describe in detail the trajectory of several key contact positions during the manual shoveling of soil in a cycle, the movement trajectory of each 45° rotation of the arm is analyzed in small stages. Among them, the black line represents the shovel, the cyan line represents the worker’s arm, the red curve represents the hand-to-shovel contact point A, the hand-to-shovel handle contact point B, and the shovel knife C point from left to right. When the arm rotation is at 0°~135°, the shovel enters the preshoveling stage; when the arm rotation is at 135°~315°, it represents the blade entering the soil. Finally, when the arm is at 315°~360°, it indicates the shovel-throwing process.

### 2.1. Crank Connecting Rod Shovel Mechanism Design

Inspired by the manual shoveling operation, the crank, shovel bar, swing bar, and bottom bracket are designed in sequence. The crank corresponds to a person’s arm, the connection between the crank and the shovel bar corresponds to the contact point between one hand and the shovel bar, the connection between the swing bar and the shovel bar corresponds to the contact point between the other hand and the shovel bar, and the bottom bracket corresponds to a person’s foot, which plays a stabilizing role.

From the perspective of ensuring the optimal combination of the working depth and its rotating shaft and the imitation of the human shoveling mechanism, the initial selection of the shovel knife length is 335 mm, the length of the shovel handle is 225 mm, the length of the crank is 140 mm, and the length of the rocker is 370 mm, which is the key size of the shoveling mechanism. At both ends of the crank, there is a positive hexagonal hole and a bearing mounting hole, which are used for installation on the output shaft of the microcultivator and the short shaft, respectively. The shovel bar has a bearing mounting hole in the shaft and a short shaft; a shovel knife is mounted at the other end of the shovel bar, and the middle pin shaft of the shovel bar is used to mount the swing rod through the bearing. The other end of the swing rod is mounted with the bottom bracket through the bearing. A three-dimensional diagram of the shoveling mechanism is shown in Figure 4. The bearings are mounted with a shoulder at one end of the shaft, which may be in contact with the inner or outer ring of the bearing, thus limiting the movement of the bearing in the axial direction.

To ensure that the crank connecting rod shoveling mechanism can be moved in the desired manner, it is necessary to analyze the relationship between its members, as shown in Figure 5. The design of the crank linkage shoveling mechanism sketch sets the lengths of the crank AB, connecting rod BC, rocker CD and stander AD to l_1_, l_2_, l_3_, and l_4_. Its length meets the requirements of the crank linkage mechanism, and the length of the connecting rod C point to the E point is l_5_. Where θ_3_ is the angle between the rod CD and the horizontal line, θ_2_ is the angle between the rod BC and the horizontal line, θ_1_ is the angle between the rod AB and the horizontal line.

The speed of the crank as the input transmission member in the entire shovel mechanism is synthesized by the speed of the machine pulling forward at a rate of 2.5 mm/s, at which the relative speed of the crank itself rotating around the shaft = 0.2 rad/s. The synthesized velocity vector equation is
(1)$$\vec{V}={\vec{V}}_{q}+{\vec{V}}_{0}$$

When the length of the crank, the angle of rotation and the speed of the machine motion are determined, the trajectory equation of the crank can be obtained, and the trajectory curve at the end of crank B can be known. As shown in Figure 6, the forward direction of the implement is positive for the x-axis, and the vertical downward direction indicates the forward direction of the y-axis.

According to the parametric equation of the circle, it can be seen that in a certain period of time t, the trajectory parametric equation at crank endpoint B can be expressed as
(2)$$\left\{ \begin{array}{l} x=V_{q}t+l_{1}\cos\omega t\\ y=l_{1}\sin\omega t \end{array}\right.$$

The connecting rod in this study will be divided into two parts. With the rocker connection as the node, the connecting rod is, in turn, divided into connecting rod BC and connecting rod CE. Connecting rod BC is connected to the power input, and the end of connecting rod CE is connected to the output—the shovel knife. First, the motion analysis of connecting rod BC is performed via MATLAB to calculate the motion trajectory of connecting rod BC, as shown in Figure 7.

Further calculation of the motion trajectory parameter equation at the connecting rod endpoint C in a certain period of time t can be expressed as
(3)$$\left\{\begin{array}{c} x_{C}=l_{3}{\mathrm{cos\theta}}_{3}+V_{q}t\;\\ y_{C}=-l_{4}+l_{3}\sin\theta_{3} \end{array}\right.$$

The trajectory equation is derived from the time t as follows:(4)$$\left\{\begin{array}{c} {x\prime }_{C}={-\omega_{3}l}_{3}{\mathrm{sin\theta}}_{3}+V_{q}\;\\ {y\prime }_{C}=-l_{4}+\omega_{3}l_{3}\cos\theta_{3} \end{array}\right.$$

During the continuous motion of the machine, the motion curve of the connecting rod BC endpoint C is a continuously biased cycloidal line, as shown in Figure 8.

Second, the motion trajectory of the connecting rod CE is analyzed, and the motion trajectory of the connecting rod CE is calculated by MATLAB, as shown in Figure 9:

Further calculated by MATLAB, the motion trajectory at the connecting rod endpoint E, as shown in Figure 10, is the motion trajectory of the connecting rod endpoint E. The motion trajectory parameter equation can be expressed as
(5)$$\left\{\begin{array}{c} x_{E}=l_{3}{\mathrm{cos\theta}}_{3}+l_{5}{\mathrm{cos\theta}}_{2}+V_{q}t\\ y_{E}=l_{3}{\mathrm{sin\theta}}_{3}+l_{5}{\mathrm{sin\theta}}_{2}\; \end{array}\right.$$

The trajectory equation is derived from time t to obtain the following:(6)$$\left\{\begin{array}{c} {x\prime }_{E}=-\omega_{3}l_{3}{\mathrm{sin\theta}}_{3}-\omega_{2}l_{5}{\mathrm{sin\theta}}_{2}+V_{q}\\ {y\prime }_{E}={\omega_{3}l}_{3}{\mathrm{sin\theta}}_{3}+\omega_{2}l_{5}{\mathrm{sin\theta}}_{2} \end{array}\right.$$

When $\theta_{2}$ = 90°, ${x\prime }_{E}$ = $-\omega_{3}l_{3}-\omega_{2}l_{5}+V_{q}<0$, which is the condition of rototilling.

For further velocity analysis of this mechanism, the closed vector equation composed of the rods and arrows of the closed graph ABCDA can be written as follows:(7)$$\vec{l_{1}}+\vec{l_{2}}=\vec{l_{3}}+\vec{l_{4}}$$

Its plural form is expressed as
(8)$$l_{1}e^{i\theta_{1}}+l_{2}e^{i\theta_{2}}=l_{3}e^{i\theta_{3}}+l_{4}e^{i\theta_{4}}$$

Equation (8) can be used to obtain the first derivative with respect to time t to obtain the following velocity relationship:(9)$$l_{1}\omega_{1}e^{i\theta_{1}}+l_{2}\omega_{2}e^{i\theta_{2}}=l_{3}\omega_{3}e^{i\theta_{3}}$$

Separating the real part and imaginary parts of Formula (9) can be carried out as follows:(10)$$\left\{ \begin{array}{l} -l_{1}\omega_{1}\cos\theta_{1}-l_{2}\omega_{2}\cos\theta_{2}=-l_{3}\omega_{3}\cos\theta_{3}\\ l_{1}\omega_{1}\sin\theta_{1}+l_{2}\omega_{2}\sin\theta_{2}=l_{3}\omega_{3}\sin\theta_{3} \end{array}\right.$$
(11)$$\left\{\begin{array}{c} \omega_{2}=\frac{{\mathrm{sin\theta}}_{1}{\mathrm{cos\theta}}_{3}-{\mathrm{cos\theta}}_{1}{\mathrm{sin\theta}}_{3}}{{\mathrm{cos\theta}}_{2}{\mathrm{sin\theta}}_{3}-{\mathrm{sin\theta}}_{2}{\mathrm{cos\theta}}_{3}}\frac{l_{1}\omega_{1}}{l_{2}}\\ \omega_{3}=\frac{{\mathrm{sin\theta}}_{1}{\mathrm{cos\theta}}_{2}-{\mathrm{cos\theta}}_{1}{\mathrm{sin\theta}}_{2}}{{\mathrm{sin\theta}}_{3}{\mathrm{cos\theta}}_{2}-{\mathrm{cos\theta}}_{3}{\mathrm{sin\theta}}_{2}}\frac{l_{1}\omega_{1}}{l_{3}} \end{array}\right.$$

When the mechanism l_1_ = 115 mm, l_2_ = 235 mm, l_3_ = 300 mm, l_4_ = 320 mm, and l_5_ = 350 mm, the angular displacement line diagram of the mechanism is finally calculated by MATLAB, as shown in Figure 11. The angular velocity line diagram is shown in Figure 12, and the angular acceleration line diagram is shown in Figure 13.

### 2.2. Simulation System Development

To further reveal the law of soil shoveling with a hand-held human-like shoveling machine, this section will use the Adams to establish a simulation model of a hand-held human-like shoveling machine and then use the ANSYS/Workbench module to simulate and analyze the key components of the shovel, providing a corresponding theoretical basis for the subsequent use of a strain gauge plate transducer.

#### 2.2.1. Establishment of the ADAMS Model

In the SW2018 software (SOLIDWORKS(R) Premium 2018 x64), the key components of the shovel are tested in multiple presimulated motion tests, and the rotary motor and related constraints are applied to the mechanism shaft to verify whether the mechanism can operate normally initially. On this basis, the stability and reliability of the mechanism are guaranteed. The model is saved as a Step format and is imported into ADAMS. The properties of the mechanism material are first defined. This is followed by the addition of a motion pair between the components, such as applying a rotation pair on the hinge while adding a moving pair between the hinge and the ground, the rotating shaft and the crank fixed constraint, the short shaft and the crank fixed constraint, and the shovel rod and the short shaft between the rotating pair. In addition, a rotation pair is added to one end of the two pendulum rods and the fixed pin shaft in the middle of the rod, a fixed constraint is applied to the shovel bar and the shovel knife, a rotation pair is applied to the other end of the two swing rods, and the base frame to apply a rotation pair and a movement pair is applied between the base frame and the ground. The corresponding rotary and translational drives are then defined on the hinge and base frame, as well as forces and moments in three directions at the center of mass of each component. The resulting ADAMS model is shown in Figure 14.

#### 2.2.2. EDEM Simulation Model Preprocessing

The soil particle shape was equated to a sphere, the radius was chosen to be 8 mm, and Poisson’s ratio was chosen to be 0.35 based on the soil (loam) texture and empirical equation. The basic parameters for the soil model obtained via the literature and soil test data are given in Table 1 [22,23,24,25].

The interaction parameters for the soil particles include the static friction coefficient, rolling friction coefficient, and recovery coefficient. The determination of this part of the parameters required a large number of experiments to be carried out, and this paper used the soil accumulation angle as the base parameter to determine this part of the interaction parameters. The particle parameters that are similar to those obtained from the soil accumulation angle test were selected in the GMEE material library of EDEM2020.2, and a virtual simulation model was established to simulate the falling process for soil in its natural state as an approximate representation of the particle interaction parameters. After several rounds of simulation tests, the interaction parameters for each material were finally determined, as shown in Table 2 [26].

According to the above tests, the soil shows a loam texture in the test area, and the soil moisture content in this area is high. Therefore, the Hertz–Mindlin with bonding model was selected as the soil particle contact model.

In the order of the model tree, soil-related parameters and basic equipment material parameters are added to the main interface particle material and its equipment material column. A single pellet sphere is selected when adding particles, and its radius and bond radius are entered to obtain a particle model, as shown in Figure 15.

The “.step” format file mentioned above is imported into the geometry, and then each component name is mapped in the geometry of the Adams model above. Here, the basic parameters of the soil particles and equipment materials are set, and then the definition of the particle factory is set according to Table 1. The upper surface is set to empty, the purpose of which is to define the upper surface as the virtual generating surface of the particles, where a dynamic form of fixed mass is generated per unit time. According to the actual working position requirements, the relative position of the soil particle bed of the geometric model is adjusted by the center coordinate. At the same time, the bonding model in the particle properties is selected, the bond generation time is defined as the particle bed after the particle bed is fully generated and is set to 0.22 s, and the remaining parameters are filled in according to the other parameters. Finally, in the simulation environment area, the position of the soil particle bed and the outer edge of the model are set to reasonable values, and the size of the outer edge of the environment should be guaranteed to be as large as possible considering the edge positions of both. To ensure the integrity of the simulation, the established model is shown in Figure 16.

#### 2.2.3. EDEM—ADAMS Coupled Simulation

According to the actual working speed and forward speed, the operating conditions of 0.2 m/s, 90 r/min, 0.3 m/s, and 140 r/min are entered into the Adams drive module, combined with the soil and metal parameters set in the EDEM above, and the coupling interface of Adams is connected with the EDEM2020.2.

During the simulation process, the motion status of the mechanism and soil can be viewed in real time in the EDEM 2020.2, and the phase process of the periodic operation of the shovel knife is shown in Figure 17. The overall operation effect is shown in Figure 18.

The simulation test results show that under the conditions of 0.2 m/s and 90 r/min, the torque of the shaft is 95.1595 N·m, and the bending moment of the shovel bar is 97.0213 N·m. At 0.3 m/s and 140 r/min, the shaft receives a torque of 76.5831 N·m, and the bending moment of the shovel bar is 82.0142 N·m. Therefore, without considering the influence of other factors, a preliminary judgment that appropriately increases the working speed and the magnitude of the rotational speed at the same time will be conducive to reducing the loads arising.

### 2.3. Tuning of the Simulation System

#### 2.3.1. Complete Machine Structure and Technical Parameters

The main structure of the human-like shoveling machine designed in this paper is shown in Figure 19, which is mainly composed of a microcultivator frame, a shoveling mechanism, and a walking mechanism. The main parameters are as follows: supporting power 4.0 kW, working speed 0.1~0.45 m/s, operating speed 65~165 r/min, working depth 90~120 mm, and working width 600 mm.

According to the shape and size of the shoveling mechanism, a reasonable support frame is selected, and the two ends of the support frame are welded to the microcultivator frame. The rotating shaft on the side of the microcultivator is fixed and installed at one end of the crank, and the shovel knife orientation is consistent with the direction of the tool movement so that the shovel knife can shovel the soil forward. Among them, the bottom bracket is fixed with the support frame to ensure the stable operation of the shoveling mechanism. On the other side of the microcultivator shaft is a small active sprocket with two bearing seats installed on both sides of the bottom end of the support frame. The walking wheel and the sprocket are installed on the bottom shaft and supported by the bearing seat, the power transmission of the walking wheel is provided by the reduction chain transmission form, and the walking mechanism is used to drive the motion of the whole machine.

After completing the structural design of the complete shovel, the physical prototype was processed, and its working effect was verified. The depth of the shovel and the stability of the shovel depth were tested, and the torque of the shaft and the bending moment of the shovel bar were investigated.

The field test was carried out in September 2021 in the outdoor soil trough of Jiangsu University, with an average moisture content of approximately 14.538% in the test area and an average soil compactness at 10~12 cm of 2037.5 kPa in a flat sandy clay loam. The soil supported a gasoline microcultivator with a power of 4 kW.

#### 2.3.2. Soil Parameter Test

The test site of this study is located in the soil bin of the agricultural side building of Jiangsu University, so the soil in the soil bin is selected as the object of research and analysis.

According to experimental soil texture measurement, the percentage of various types of soil particles contained in the soil is accurately obtained and matched with the international soil texture classification standard to further analyze the properties of the soil [27]. The experiment adopted soil sieves with a diameter of 20 cm and pore diameters of 2 mm, 1.5 mm, 0.1 mm, 0.075 mm, and 0.05 mm. A known quantity of dry crushed soil was used according to the three tests for hierarchical screening, and the quality of the soil particles left in each of the sieves was recorded.

Experimental data show that the average content of powder clay particles in the soil is 36.88%, and the average content of sand particles is 63.12%. Thus, the soil is determined to be sandy clay loam according to the international soil texture classification standards.

The soil moisture content experiment not only intuitively reflects the wetness and tightness of the soil but also provides a theoretical basis for the soil bond radius in the simulation analysis below. This experiment used the drying method to test the moisture content. Ten aluminum boxes with uniform and known quality were placed in an oven for ten hours at a temperature of 105 °C until the water in the soil was completely evaporated. The moisture content of the soil was calculated by measuring the mass difference before and after drying. The experimental data showed that the average moisture content in the measured soil was 14.538%.

In this paper, the JDM-1 electric relative density meter is used to test the soil density, which provides an accurate theoretical basis for the soil model parameters in the simulation below, thereby improving the reliability of the coupled simulation. The experimental data showed that the average soil density was 2.5291 g/cm^3^.

A ZJ-type strain-controlled straight shear instrument was used to carry out a straight shear test on the soil sample, and vertical stresses of 100 kPa, 200 kPa, 300 kPa, and 400 kPa were applied to the soil sample three times to obtain the maximum average shear strength, internal friction angle, static friction coefficient, and adhesion force. Experimental data show that the average adhesion force is Ca.A = 24.41 kPa, the internal friction angle is φ = 31.62°, and the static friction coefficient is K = 0.61576.

The maximum compressive strength of the soil was tested by a YYW-2 strain-controlled unattended pressure gauge, which provided a theoretical basis for the soil parameters in the subsequent simulation. The experimental results showed that the average axial strain was 3.75%, the axial displacement difference was 3 mm, and the principal stress difference was 122,494.64 kPa.

#### 2.3.3. Establishment of Soil Models

To ensure the reliability of the simulation, this study set the soil area length to 3500 mm, the width to 900 mm, the thickness of the soil layer to 200 mm, and the single soil particles as small spheres, considering that with the size of the particle size in the discrete element simulation and the simulation operation time being inversely proportional, too small a particle diameter will directly lead to a high rate of increase in the simulation time [28]. Therefore, the particle radius is 8 mm in this study. According to the soil particle size analysis experiment, The Hertz–Mindlin model with bonding was selected in the EDEM 2020.2 as a contact model between the soil particles. The bond radius is calculated according to the soil moisture content experiment [29], $W_{s}=\frac{\rho_{s}\cdot V_{s}}{\rho_{k}\cdot V_{k}},V_{k}=\frac{4}{3}\pi R_{k}^{3},V_{s}=\frac{4}{3}\pi \left(R_{n}^{3}-R_{k}^{3}\right)$. The bond radius is obtained as $R_{n}$ = 8.554 mm based on soil density tests $\rho_{k}$ = 1529.1 kg/m^3^. The soil Poisson ratio is based on the formula [30] $\mu =\frac{K_{\varphi }}{1+K_{\varphi }}$, and $K_{\varphi }=1-\sin\varphi$, where φ is the internal friction angle sought in the straight shear test and a value of μ = 0.32 is obtained. Among them, the soil shear modulus is obtained according to the calculation formula [31] G = E/2(1 + μ), and the shear modulus G = 1237319.567 Pa is obtained. The soil accumulation angle experimental data were imported into the GEMM database in EDEM to obtain the corresponding soil recovery coefficient, static friction coefficient, and dynamic friction coefficient [32]. Unit normal stiffness, critical method stress, unit tangential stiffness, and critical tangent stress are obtained using the unconfined compression test. Based on the relevant literature [33] on the contact parameters between soil and steel, the relevant parameters of the soil and steel model are determined as shown in Table 3.

#### 2.3.4. Implement Load Test System

The strain gauge is directly attached to the metal surface to react to the load situation via deformation. Thus, to ensure the accuracy of the testing process, it is necessary to perform a static analysis of the relevant parts before pasting the strain gauge. This study uses the ANSYS/Workbench module to simulate and analyze the key components of the scraper, providing a corresponding theoretical basis for the subsequent use of the strain gauge plate transducer.

To improve the simulation efficiency, SolidWorks saved the established 3D model as a (step) format file into The Workbench, set the material parameters of the metal according to Q235 steel, divided the mesh according to the appropriate size, set fixed constraints on the base frame and one end of the shaft, applied a torque of 1000 N·m on the rotating shaft, and set a force of 50,000 Pa on the shovel knife to ensure that the part is more obvious under a sufficiently large load. The shaft deformation in Figure 20 is shown after torque is applied and after the simulation is completed, and the deformation of the shovel rod with applied force is shown in Figure 21.

According to the simulation results, the variation in shape near the crank end position on the rotating shaft is the largest, and the variation in shape is the largest near the shovel knife position on the shovel bar. Considering the limitations of its space, the strain gauge is pasted in a reasonable location according to the simulation results.

The materials used in applying the strain gauge mainly include the strain gauge itself, absolute ethanol, cotton, coarse/fine sandpaper, a marker pen, a small leather ruler, quick-drying glue, insulating tape, wiring, solder, and a multimeter. After polishing the rust and impurities of the metal surface with coarse sandpaper to brightness, fine sandpaper was used to polish it in one direction. The surface was wiped with cotton soaked in absolute ethanol until the cotton did not blacken. Then, after the surface was dried with dry cotton, an appropriate amount of quick-drying glue was dropped on the reverse side of the strain gauge. A total of four strain gauges were immediately pasted onto the metal surface, and four strain gauges were pasted at 90° intervals around the axis of the shaft and the shovel rod in the same way. Insulation tape was wrapped around the strain gauge and the component to which it was attached. The wires were soldered to the strain gauge, and the wire and strain gauge were tested with a multimeter to detect circuit errors. The strain gauge and wiring were then connected with the other original modules. Finally, the entire test system was protected, as shown in Figure 22.

To avoid systematic error caused by the different performances of the strain gauges, the same batch of resistance strain gauges with a resistance value of 350 Ω is used as the bridge arm of the bridge, and the appropriate position is selected as the chip point according to the above simulation analysis results and the actual operation of the tool. The strain gauges are grouped according to the layout of the whole bridge, as shown in Figure 23.

To ensure the reliability of the measurement results, the applied torque and bending moment strain gauges are calibrated before the field load test. The equipment required for torque calibration is a 5 V lithium battery, a wrench, several weights of different magnitudes, and an oscilloscope. The equipment required for bending moment calibration are 5 V lithium batteries, wires, weights, and oscilloscopes, and the calibration process is shown in Figure 24.

According to the fitting analysis of the measured data results and the simulation data, the shaft torque and the bending moment of the shovel bar under the conditions of 0.2 m/s and 90 r/min are exported in the Adams postprocessing interface after the simulation is completed, as shown in Figure 25 and Figure 26; the shaft torque and the bending moment of the shovel rod at 0.3 m/s and 140 r/min are shown in Figure 27 and Figure 28, respectively.

## 3. Simulation System Verification Experiment

A stable working area is selected for the load test. The torque test circuit and the bending moment test circuit are connected and receive their load values at the same time via two computer equipment components. The test process is shown in Figure 29. The test was carried out under the following conditions: 0.2 m/s and 90 r/min and 0.3 m/s and 140 r/min.

### 3.1. Load Test Results

According to the field test results, the torque value of the tool at an operating speed of 0.2 m/s and 90 r/min is 101.1829 N·m. At 0.3 m/s and 140 r/min, the torque value is 81.1376 N·m, and the torque of the rotating shaft is shown in Figure 30 and Figure 31.

According to the above method, the torque value of the shovel bar under the conditions of 0.2 m/s and 90 r/min is 81.1376 N·m. For 0.3 m/s and 140 r/min, the torque value is 86.7668 N·m, and the bending moment of the shovel is obtained, as shown in Figure 32 and Figure 33, by filtering out some errors and sudden anomalies.

The simulation results and the field test data show that the actual measured torque and its bending moment value are slightly larger than in the simulation. The main reason for the difference between the simulation and the actual measurement is the uneven surface of the soil bin, the instability of the control, and the partial friction between the test prototype assembly and other factors described in [34,35], such as the average torque error of the rotating shaft of 6.26%, and an average error of the bending moment of the shovel of 5.43%.

### 3.2. Stability Test of Operating Effect

After the operation was completed, 20 points were randomly selected in the forward direction of the machine to test the shovel depth and width. In the test, the main instruments to be used are two straight steel rulers. At the test points, firstly, one straight steel ruler is inserted into the bottom of the shoveled soil, and then the other straight steel ruler is leveled with the ground surface. At the measurement point, the first straight steel ruler is inserted into the bottom of the shoveled soil, and then the other straight steel ruler is used to keep level with the ground surface, and the shovel depth and shovel width are measured according to the scale of the two straight rulers. This is shown in Figure 34.

At the end of the test trials, the values of shovel depth and shovel width were obtained for the machine under the optimized conditions, as shown in Table 4.

The 20 sets of shovel depth and shovel width values in Table 4. were sequentially averaged using the correlation formulae, resulting in an average shovel depth is 102.66 mm and an average shovel width is 193.5 mm, respectively.

Based on the shovel depth and shovel width test results and the average shovel depth and shovel width values above, the standard deviation of the shovel depth is 2.1228 mm, and the standard deviation of the shovel width is 11.9937 mm.

Finally, the coefficients of variation of shovel depth and shovel width were obtained as 2.0678 and 6.1983, respectively, and the coefficients of stability of shovel depth and shovel width were obtained as 97.9322% and 93.8017%, respectively.

### 3.3. Virtual Mixing Uniform Design Experiment

Uniform experimental designs can perform multifactor, multilevel experiments with a smaller number of trials [36]. To study the influence of the machine’s forward speed, output shaft speed, crank, shovel knife, and shovel knife shape on the output shaft torque and bending moment of the shovel bar during the operation of the shovel, the uniform design test of the mixing level is shown in Table 5.

According to the uniform test arrangement, a total of eight sets of simulation tests were carried out. After the coupled simulation solution is completed, the simulation result data are analyzed by importing the .res file in ADAMS. The torque of the rotating shaft and the bending torque data of the shovel bar are obtained via the postprocessing analysis interface. The maximum torque and maximum bending moment in each test are taken as the analysis objects, and the results are summarized as shown in Table 6.

In the experiment, there are quantitative factors, but there are also qualitative factors. The ordinary regression analysis of the experiment is not entirely reasonable. If the four levels of qualitative factors are proposed to be 0–1, the use of estimated quadratic terms and the interaction method for its stepwise regression analysis will be prone to the analysis of the leakage of the combination phenomenon, leading to the inability to distinguish which factors between the quadratic terms or the combination of interaction terms can generate the optimal results of the test, thereby increasing the complexity of the test and the inaccuracy of the data. Therefore, to avoid the above problems, the experimental results will be analyzed stepwise with categorical variables in the data processing system (DPS) [37,38].

The names of the blade shapes in Table 6 are further defined as dummies A4, A2, A3, A1, A1, A3, A2, and A4 in order, with the dummies placed at the far left of the list during analysis, followed by the remaining arguments in the middle and the dependent variable at the far-right end of the list. To discuss the influence of independent variables on the torque value and bending moment value separately, the torque value is first analyzed as the dependent variable.

As seen from the results in Table 7, the larger the F value is, the smaller the *p* value, indicating that the regression equation has a better significance level. The regression analysis results between the torque value and the independent variable are obtained, as shown in Table 8.

Regression models based on quadratic polynomials are
(12)$$y=\beta_{0}+\sum_{i-1}^{m}\beta_{i}x_{i}+\sum_{1\leq i\leq j\leq m}\beta_{ij}x_{i}x_{j}+\varepsilon$$

From the above corresponding *p* value, it can be seen that the magnitude of the impact on the regression model value is X3 × X4 > A2 > A1 > X4 × X4 > X1 × X4 > A3, and at the same time, the correlation coefficient is adjusted in the output results. The statistical test F value is 56,461.3503, which is much greater than the significant horizontal critical value [101], and the *p* value is 0.0032, which is less than 0.05, which shows that the correlation and goodness of fit of the model are very high. Based on the correlation coefficients listed above, the binding Equation (13) yields the corresponding regression equation as follows:(13)$$\begin{array}{l} y_{1}=-378.8148-54.5744\cdot A_{1}-89.3959\cdot A_{2}-13.95364\cdot A_{3}-0.350879\cdot X_{1}\cdot X_{4}\;+\;\\ 0.02060835\cdot X_{3}\cdot X_{4}-0.00333891X_{4}^{2} \end{array}$$

According to Formula (13), it is necessary to ensure that the regression equation has a minimum value. Therefore, it can be known by using the method of separating variable groups that there is no interaction between A1, A2, A3, A4, and other variables. To ensure that Y can produce the minimum value in this item, the virtual independent variables are separated from other independent variables, so A2 should be taken as 1, and A1, A3, and A4 should all be taken as 0. In addition, X1 is separated from X2 and X3, and the torque value is negatively correlated with the forward speed. Therefore, taking the maximum value of X1 as 0.45 can ensure that the interaction term between X1 and X4 has the minimum value. Similarly, X3 is separated from X1 and X2, and the torque value is positively correlated with the crank length. Therefore, taking X3 as 155 can ensure that the interaction term between X3 and X4 has the minimum value. At this time, only the univariate quadratic form of X4 is left in the polynomial. Because the coefficient before the quadratic term is less than 0, if the extreme point is taken, the whole equation will have a maximum value, the symmetry point is located at the right end of the maximum value of X4, and X4 is a monotonically increasing function with values between 185 and 230, so there is a minimum value when X4 is taken as 185 in the interval. However, the correlation between machine speed and torque value has not been introduced in the above analysis, so the regression analysis of the bending moment of the shovel bar is carried out to draw a conclusion.

According to the requirements, the bending moment of the connecting rod is then analyzed in the DPS operating system using the same method as in the torque value regression analysis described above.

After the stepwise regression analysis of the same categorical variables is completed, the corresponding parameters results are obtained along with the analysis of variance between the related factors, as shown in Table 9.

At the same time, the regression analysis results between the bending moment value of the shovel bar and the relevant independent variables are obtained, as shown in Table 10.

From the above corresponding *p* value, it can be seen that the magnitude of the influence on the regression model value is X3 × X4 > A2 > A1 > X1 × X2 > A3 > X3. At the same time, the correlation coefficient R = 0.99998, the adjusted correlation coefficient Rad = 0.99984, the F-test statistical value is F = 3601.3215, which is much greater than the significant horizontal critical value $F_{\left(6,1\right)}\left(0.05\right)=234$, and *p* = 0.0128, which is less than 0.05. This shows that the correlation and goodness of fit of the model are high. From the correlation coefficient binding Formula (12) listed above, the corresponding regression equation can be obtained as follows:(14)$$y_{2}=-256.7248-61.5872\cdot A_{1}-104.4889{\cdot A}_{2}-10.22914{\cdot A}_{3}-0.0907731{\cdot X}_{3}\;-0.388606\cdot X_{1}\cdot X_{2}+0.01383171\cdot X_{3}\cdot X_{4}$$

According to Equation (14), it is necessary to ensure that the regression equation has a minimum value, and the method of separating variable groups shows that there is no interaction between A1, A2, A3, A4, and other items. To ensure that y can produce the minimum value in this field, it is necessary for A2 to be taken as 1, and A1, A3, and A4 to all be taken as 0. In addition, X1, X2 and X3, X4 are separated, and the size of the bending moment value is negatively correlated with the machine’s forward speed and its absolute speed, so taking the maximum value of X1 to be 0.45, the maximum value of X2 is 140. One can thus ensure that the interaction term between X1 and X2 has a minimum value. Then, the influence of the value of X4 on the regression model is discussed because the coefficient on the X3 and X4 interaction terms is positive, and the X4 and X1, X2 are separated, so when X4 takes the minimum value, it can be guaranteed that this interaction term has a minimum value. At this time, the coefficient on X3 × X4 is 2.55886635, which is much greater than the absolute value of the X3 coefficient, so in order to ensure that the equation has a minimum value, the value of X3 is set to 155.

According to the comprehensive analysis of the above two, when the forward speed of the machine is 0.45 m/s, the shaft speed is 140 r/min, the crank length is 155 mm, the shovel knife length is 185 mm, and the sharp angle straight shovel operation is selected, there will be a minimum torque and bending moment.

## 4. Conclusions

This paper presents the design of a human-like field shovel, and the conclusions of the study are as follows:(1)Based on the principle of manual shoveling, a compact and low-cost working tool is designed. This tool reduces soil adhesion, avoids the direction of shoveling, and achieves low-power operation of the machine. Via the analysis of the existing conditions, the relative position and characteristics of the mechanism, the motion trajectory, and the motion curve of the hand-held imitation human shovel are further optimized.(2)Using discrete elements and Adams2020 kinematics simulation to establish a coupled simulation model for the key mechanism of the shovel, the simulation results show that under the conditions of 0.2 m/s and 90 r/min, the torque of the rotating shaft is 95.15958 N·m, and the bending moment of the shovel bar is 97.0213 N; the torque of the rotating shaft is 76.5831 N·m under the conditions of 0.3 m/s and 140 r/min, and the bending torque of the shovel bar is 82.0142 N·m.(3)The hand-held imitation human field shoveling machine was processed and manufactured via finite element static analysis of the crank connecting rod shoveling mechanism to determine the appropriate strain gauge paste position. For the values of the microcultivator rotating shaft torque value and the bending moment on the shovel bar, a field verification test was carried out. The experimental results show that the actual cultivation depth in the field of the prototype is approximately 10.6 cm, and the stability coefficient of the tillage depth is up to 98.15%. The rotating shaft torque and the bending moment of the shovel bar under the conditions of 0.2 m/s and 90 r/min are 101.1829 N·m and 101.9465 N·m, respectively, and the torque of the rotating shaft and the bending torque of the shovel bar are 81.1376 N·m and 86.7668 N·m, respectively, under the conditions of 0.3 m/s and 140 r/min.(4)Due to the unevenness of the soil surface in the field and the instability of the operation, the verification test shows that the average torque error of the rotating shaft is approximately 6.1384%, and the average error of the bending moment of the shovel bar is 5.4356%. According to the analysis, the values of both are within the theoretical error, which proves that the simulation and the actual operation are consistent. The minimum power during operation is 1.275 kW. The main reasons for the difference between the simulation and actual measurement are two-fold. For field measurement, this difference is due to the uneven surface of the soil bin, the instability of the control, the partial friction between the test prototype assembly, and other factors. For the simulation experiment, due to the difference between the soil particle model and the actual situation, the microscopic parameters, such as the normal and tangential stiffness of the soil, cannot be accurately measured by the soil test. Second, in the existing knowledge field, the basic correlation analysis can only be carried out by the discrete element system.(5)A uniform test is designed, and the coupled simulation analysis of the operating process of the shovel mechanism is carried out by using the discrete element method and ADAMS2020. By regression analysis of the test results, the mathematical model of the relationship between the forward speed of the machine, the rotating speed of the rotating shaft, the crank length, the shovel knife length, and the shovel knife shape corresponding to the minimum bending moment of the shovel bar is obtained, and the optimal working parameters are obtained according to this model.

## Figures and Tables

**Figure 1 sensors-24-00868-f001:**
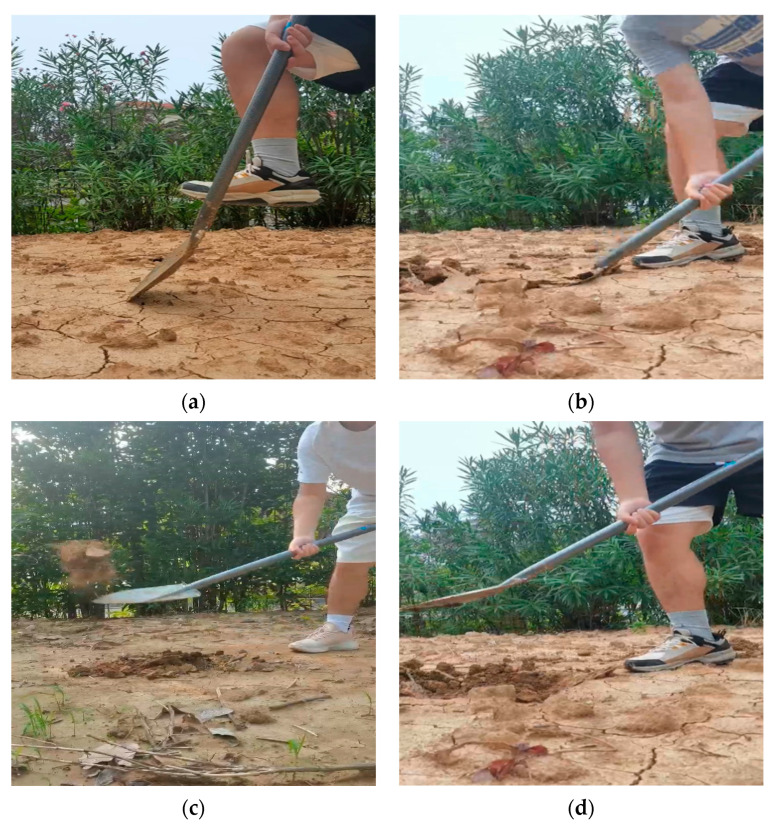
Principle of manual operation. (**a**) The process of entering soil; (**b**) the process of shoveling soil; (**c**) the process of throwing soil; (**d**) the process of resetting.

**Figure 2 sensors-24-00868-f002:**
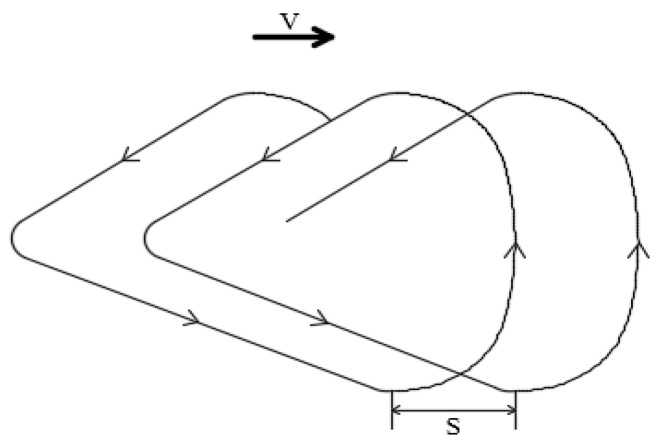
Shovel knife movement trajectory.

**Figure 3 sensors-24-00868-f003:**
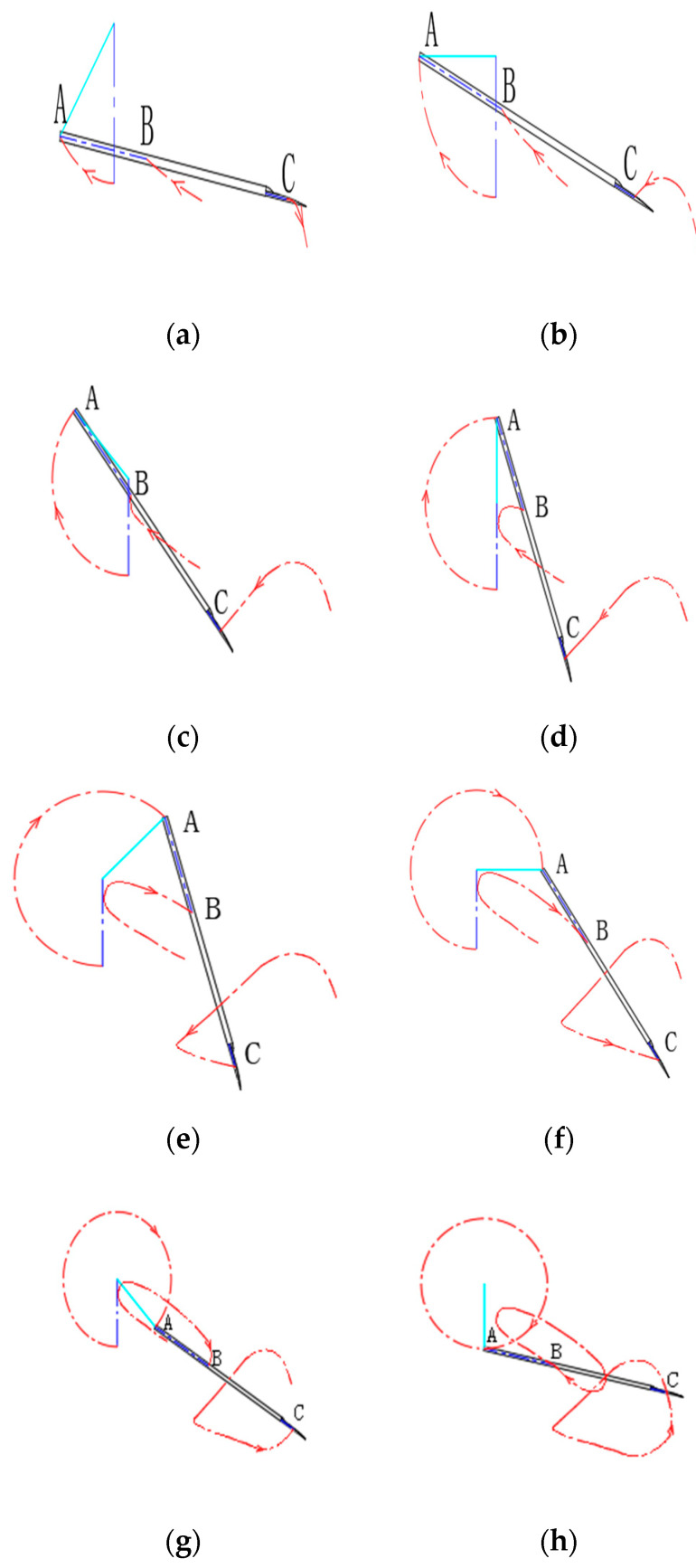
Curve of manual soil shoveling. (**a**) Rotate 45°; (**b**) rotate 90°; (**c**) rotate 135°; (**d**) rotate 180°; (**e**) rotate 225°; (**f**) rotate 270°; (**g**) rotate 315° (**h**) rotate 360°.

**Figure 4 sensors-24-00868-f004:**
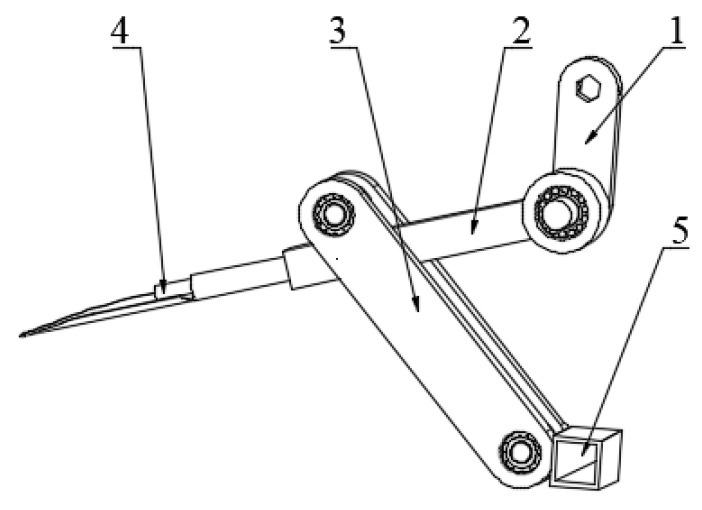
Assembly of the key parts of the shoveling machine. 1—crank, 2—link, 3—rocker, 4—Schleifer, 5—rack.

**Figure 5 sensors-24-00868-f005:**
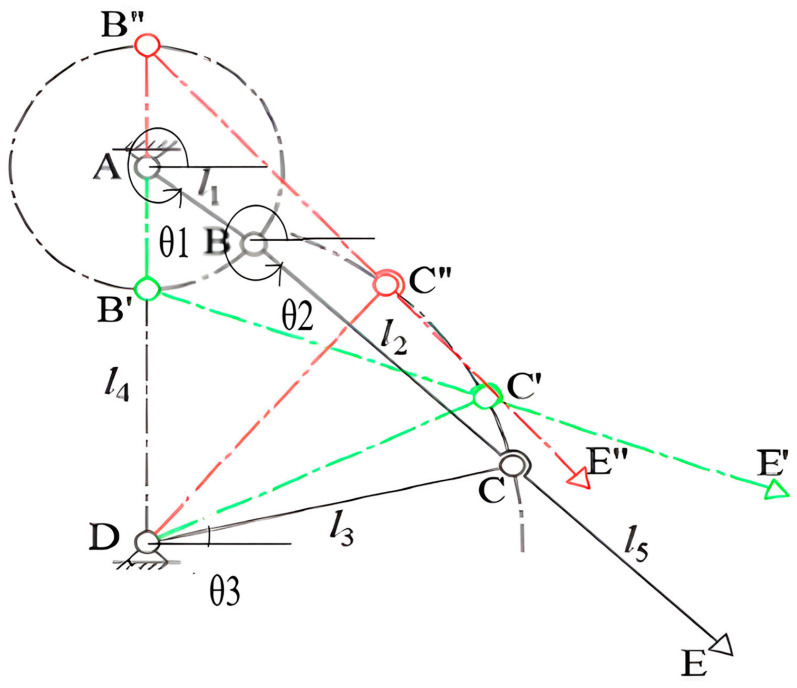
Crank link soil shovel mechanism.

**Figure 6 sensors-24-00868-f006:**
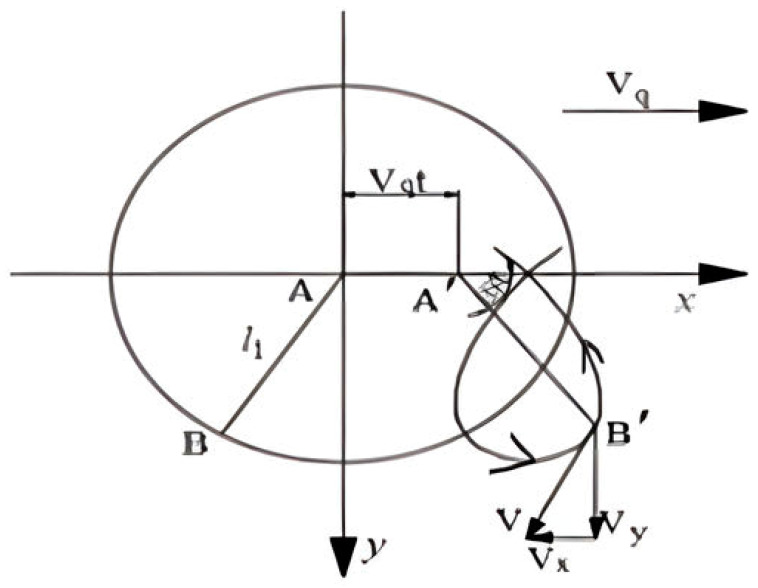
The motion trajectory of the crank.

**Figure 7 sensors-24-00868-f007:**
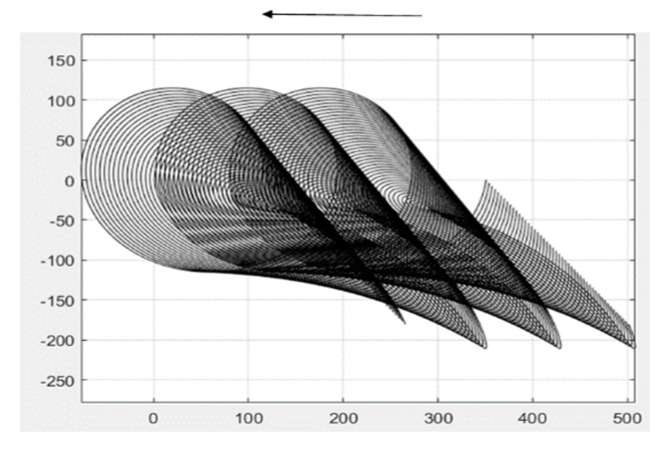
The motion trajectory of linkage rod BC.

**Figure 8 sensors-24-00868-f008:**
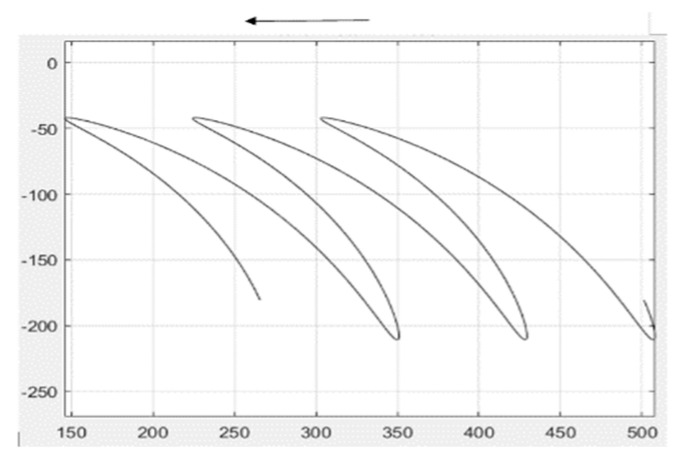
The motion curve of linkage rod BC.

**Figure 9 sensors-24-00868-f009:**
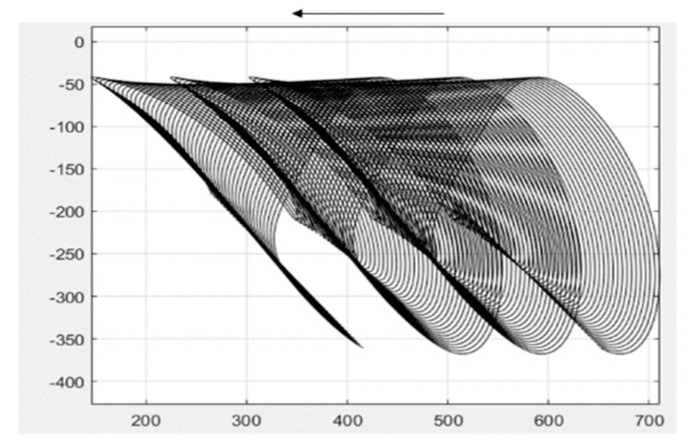
The motion trajectory of linkage rod CE.

**Figure 10 sensors-24-00868-f010:**
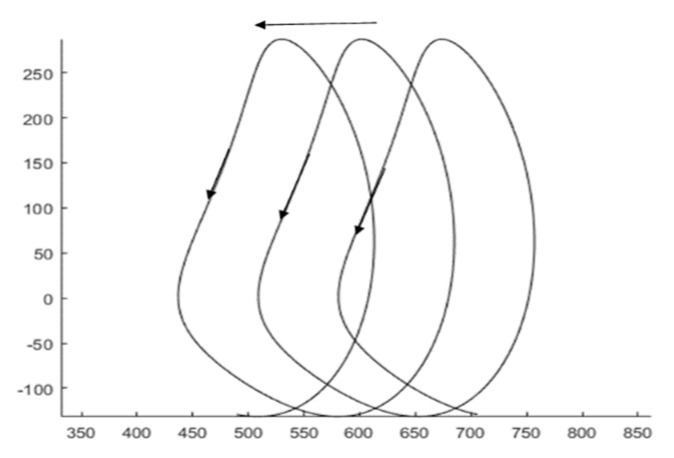
The movement curve at linkage rod E.

**Figure 11 sensors-24-00868-f011:**
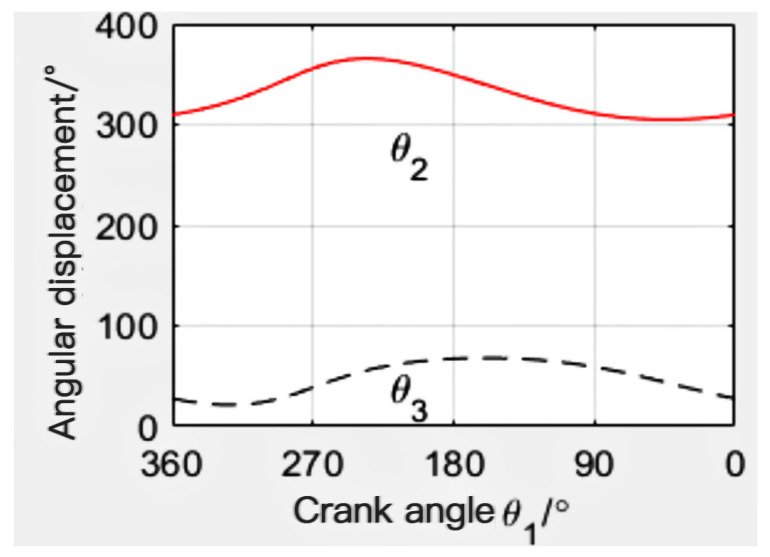
Angular displacement diagram.

**Figure 12 sensors-24-00868-f012:**
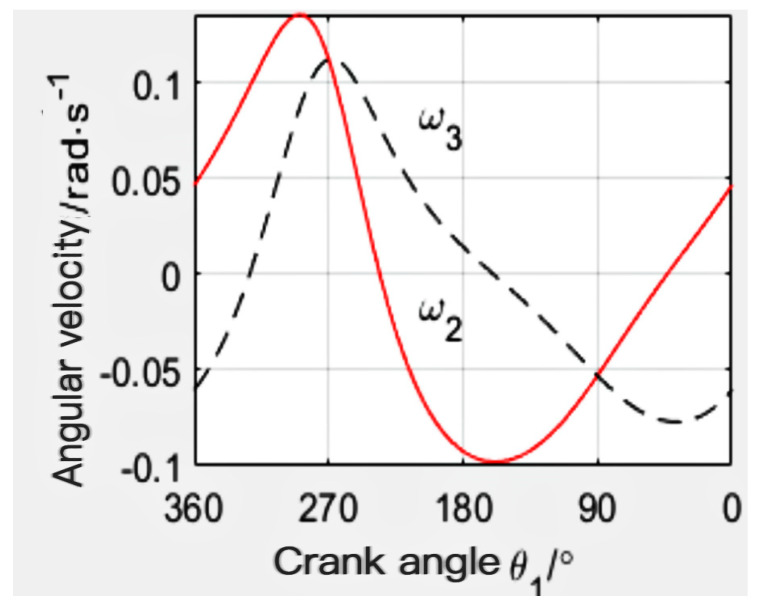
Angular velocity diagram.

**Figure 13 sensors-24-00868-f013:**
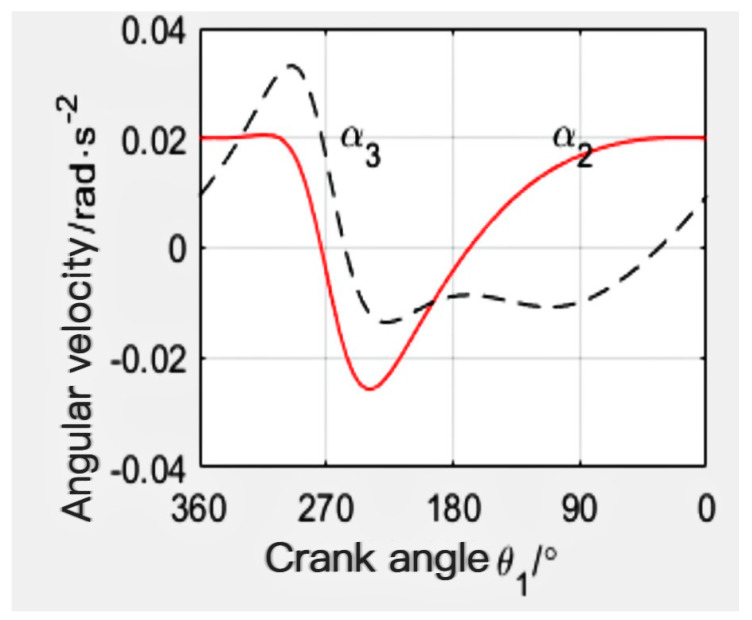
Angular acceleration diagram.

**Figure 14 sensors-24-00868-f014:**
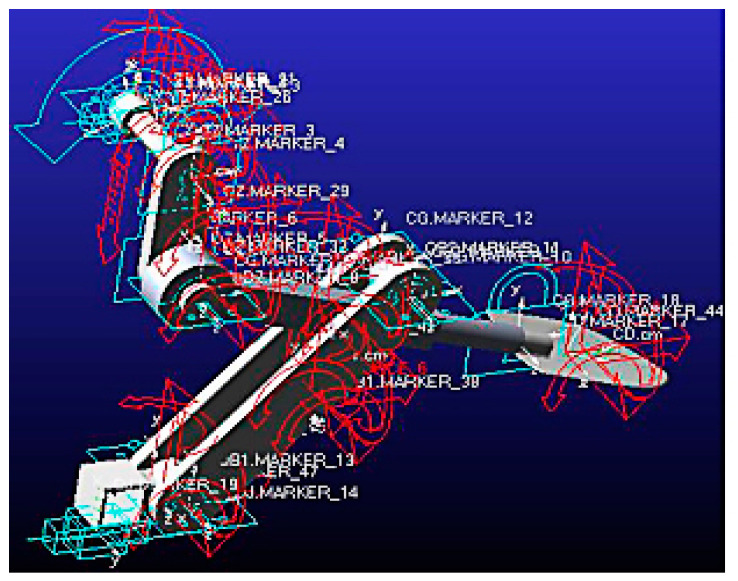
The ADAMS simulation model.

**Figure 15 sensors-24-00868-f015:**
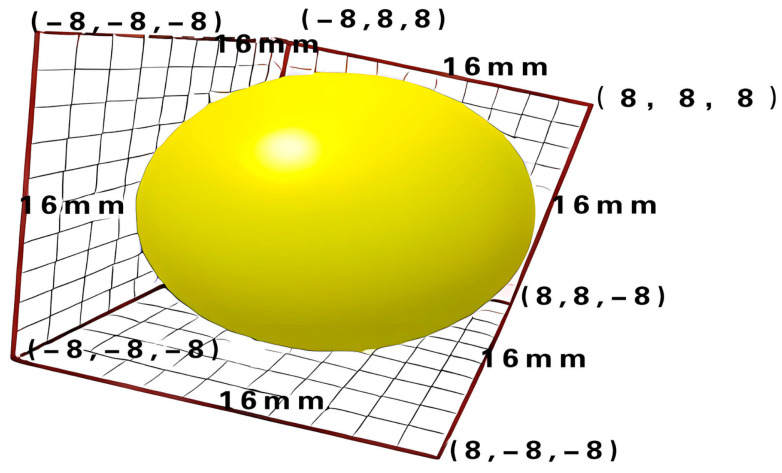
Single particle model.

**Figure 16 sensors-24-00868-f016:**
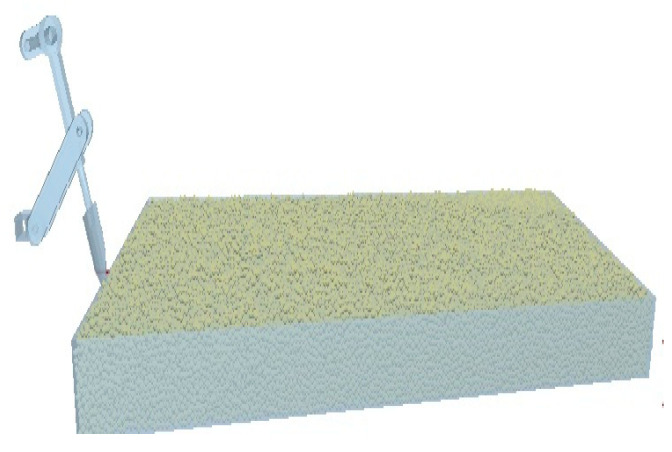
Simulation model.

**Figure 17 sensors-24-00868-f017:**
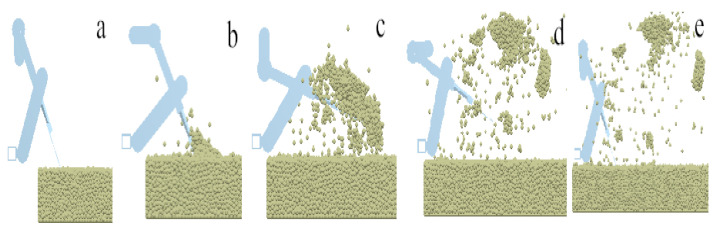
Simulated operating process of the shovel knife. Point a is the point where the crank is connected to the frame, point b is the point where the crank is connected to the connecting rod, point c is the point where the rocker is connected to the connecting rod, point d is the point where the frame is connected to the rocker, and point e is the point where the shovel rod ends.

**Figure 18 sensors-24-00868-f018:**
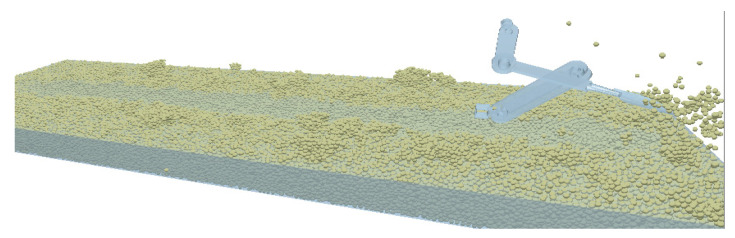
Simulation effect.

**Figure 19 sensors-24-00868-f019:**
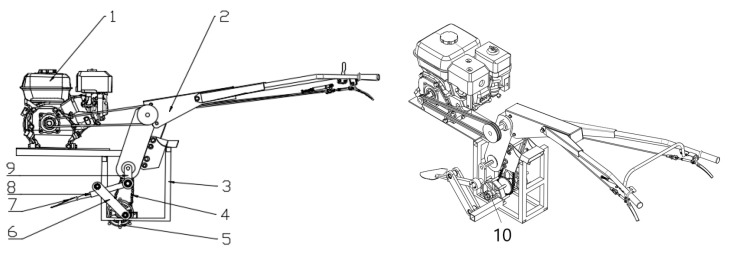
An imitation of the human field weeding machine. 1—gasoline engine, 2—micro-tiller machine rack, 3—support frame, 4—chain drive mechanism, 5—walking wheel, 6—rocker, 7—shovel blade, 8—link, 9—crank, 10—bearing seat.

**Figure 20 sensors-24-00868-f020:**
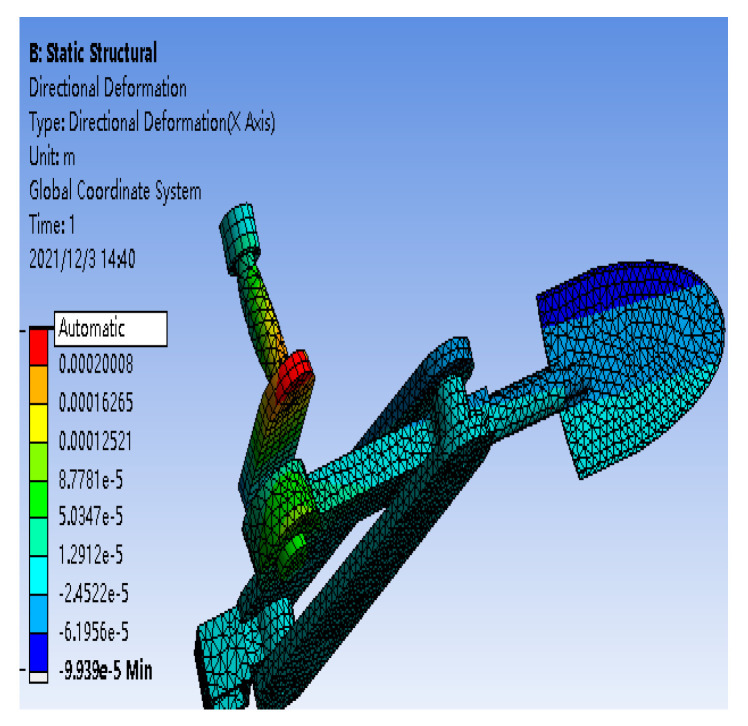
Change in the surface shape of the rotating shaft.

**Figure 21 sensors-24-00868-f021:**
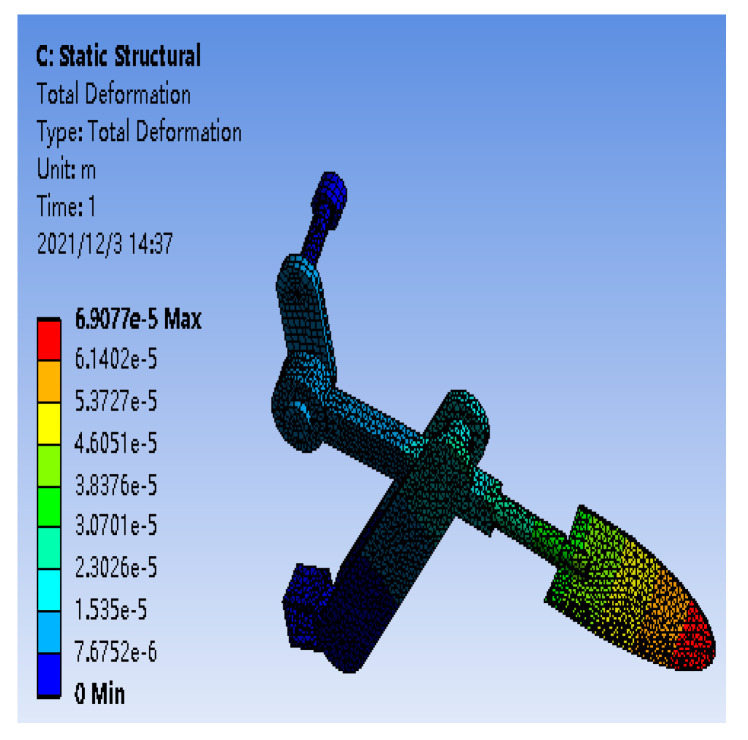
Change in the surface shape of the shovel rod.

**Figure 22 sensors-24-00868-f022:**
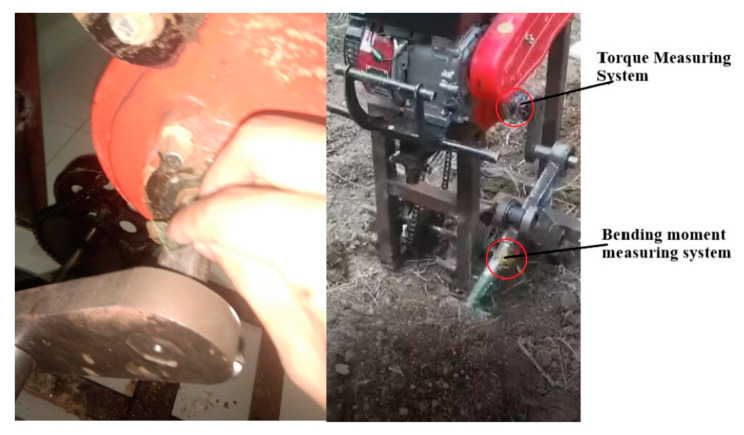
Strain sheet patch processing.

**Figure 23 sensors-24-00868-f023:**
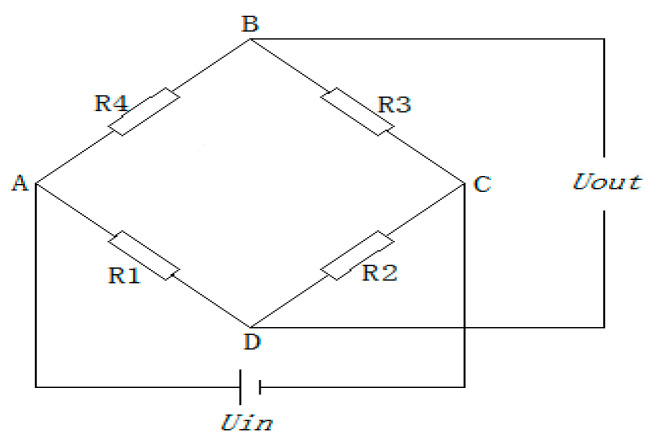
Bridge from the strain gauge.

**Figure 24 sensors-24-00868-f024:**
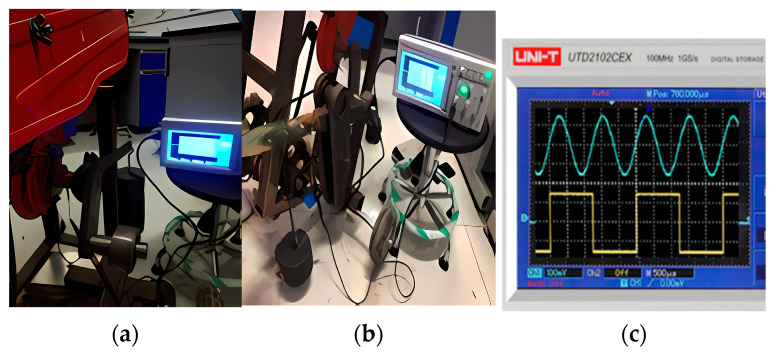
The calibration process. (**a**) Torque calibration process; (**b**) bending moment calibration process; (**c**) oscilloscope display diagram.

**Figure 25 sensors-24-00868-f025:**
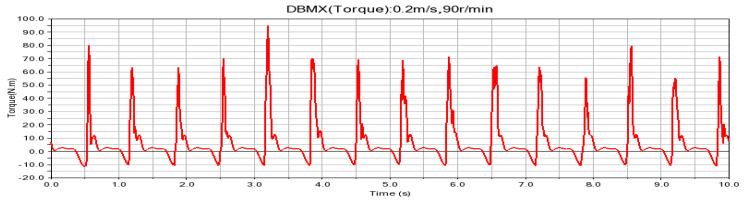
Torque on the shaft at 0.2 m/s and 90 r/min.

**Figure 26 sensors-24-00868-f026:**
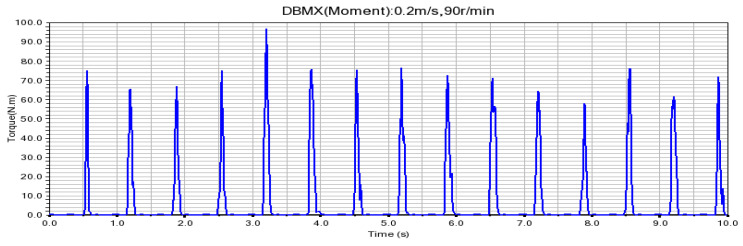
Shearing force of the shovel bar at 0.2 m/s and 90 r/min.

**Figure 27 sensors-24-00868-f027:**
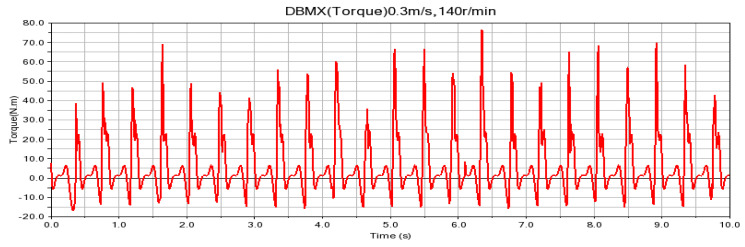
Torque on the shaft at 0.3 m/s and 140 r/min.

**Figure 28 sensors-24-00868-f028:**
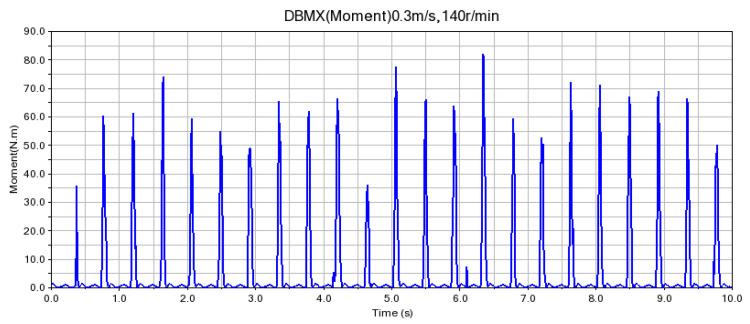
Shearing force of the shovel bar at 0.3 m/s and 140 r/min.

**Figure 29 sensors-24-00868-f029:**
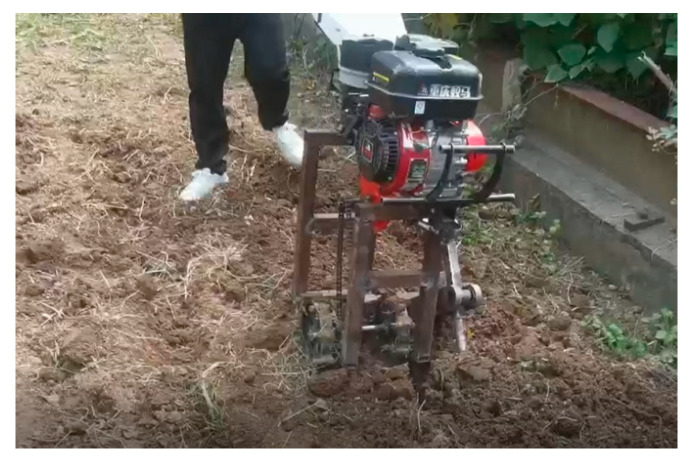
Test of the operating process.

**Figure 30 sensors-24-00868-f030:**
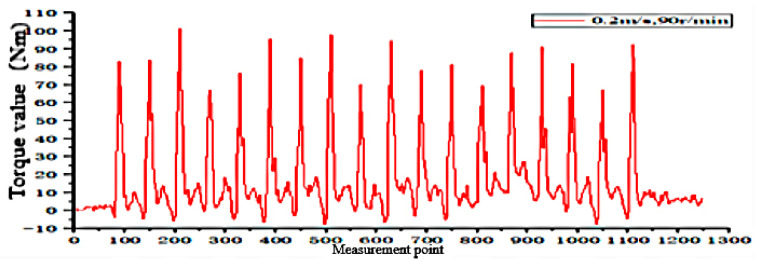
Torque value operating at 0.2 m/s and 90 r/min.

**Figure 31 sensors-24-00868-f031:**
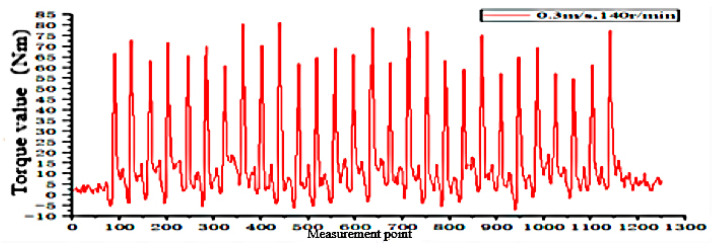
Torque value operating at 0.3 m/s and 140 r/min.

**Figure 32 sensors-24-00868-f032:**
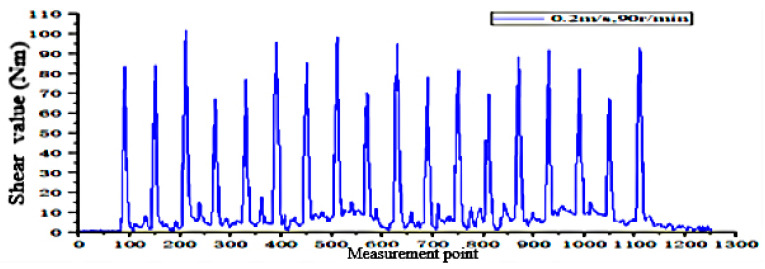
Shear value operating at 0.2 m/s and 90 r/min.

**Figure 33 sensors-24-00868-f033:**
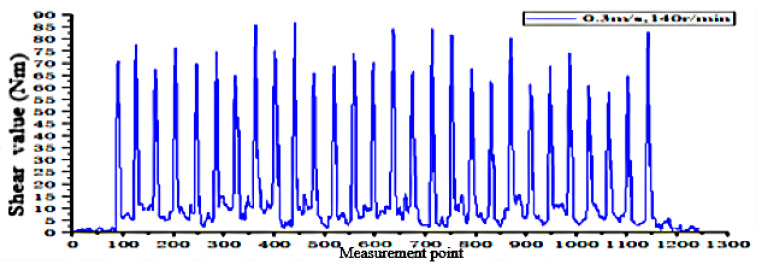
Shear value operating at 0.3 m/s and 140 r/min.

**Figure 34 sensors-24-00868-f034:**
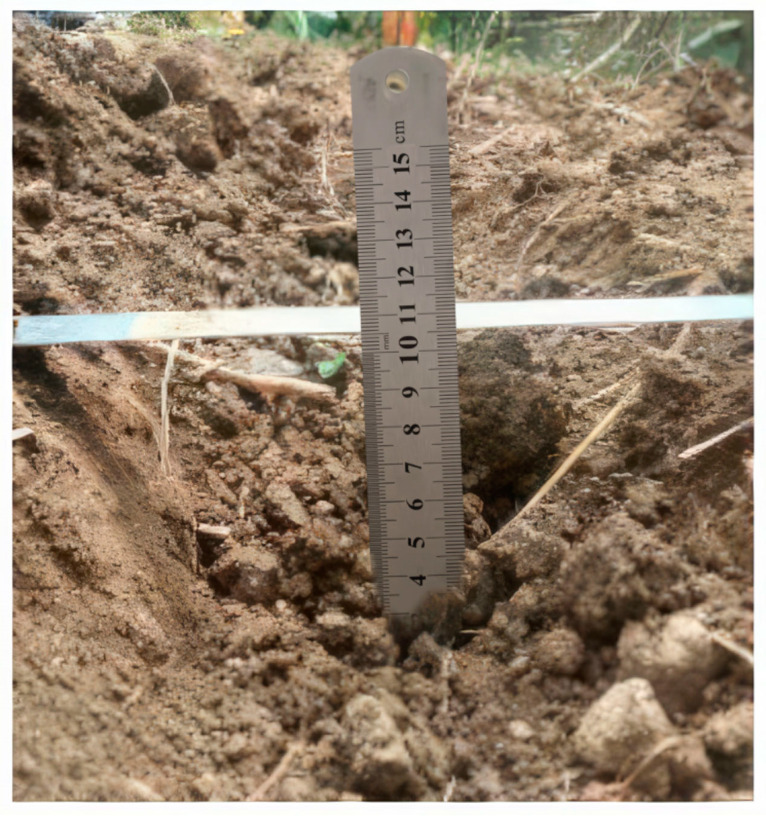
Measuring process of shovel depth and shovel width.

**Table 1 sensors-24-00868-t001:** Basic parameters for the soil simulation.

	Poisson’s Ratio	Density/kg/m^3^	Shear Modulus/pa	Particle Radius/mm
Soil	0.35	1732	1.1 × 10^6^	8

**Table 2 sensors-24-00868-t002:** Interaction parameters for soil particles.

	Recovery Coefficient	Static Friction Coefficient	Rolling Friction Coefficient
Soil	0.4	0.6	0.15

**Table 3 sensors-24-00868-t003:** Relevant parameters of the soil model.

Parameter	Numeric Value	Parameter	Numeric Value
Soil type	sandy clay loam	Normal stiffness of soil particle bonding/Pa·m^−1^	4.08 × 10^7^
Soil sample size (length × width × height)/(mm × mm × mm)	3500 × 900 × 200	Tangential stiffness of soil particle bonding/Pa·m^−1^	1.71 × 10^7^
Poisson’s ratio of soil particles	0.32	Soil particle bonding normal stress/Pa	1.225 × 10^5^
Soil particle density/(kg/m^3^)	1529.1	Soil particles bond tangential stress/Pa	8.64 × 10^4^
Shear modulus of soil particles/Pa	1.24 × 106	Poisson’s ratio of steel material	0.29
Recovery coefficient between soils	0.5	Steel material density/(kg/m^3^)	7801
Coefficient of static friction between soils	0.6	Young’s modulus of steel/Pa	2.06 × 10^11^
Coefficient of dynamic friction between soils	0.15	Recovery coefficient between soil and steel	0.3
Soil particle radius/mm	8	Coefficient of static friction between soil and steel	0.6
Binding radius of soil particles/mm	8.554	Dynamic friction coefficient between soil and steel	0.12

**Table 4 sensors-24-00868-t004:** Shovel depth and shovel width values.

Measurement Point	Shovel Depth (mm)	Shovel Width (mm)	Measurement Point	Shovel Depth (mm)	Shovel Width (mm)
1	102.5	178	11	101.7	189
2	104.3	185	12	102.4	214
3	104.9	174	13	105.3	203
4	103.2	199	14	100.3	217
5	99.1	191	15	103.1	185
6	101.5	186	16	101.6	196
7	101.8	202	17	101.4	186
8	103.1	207	18	107.5	197
9	99.4	194	19	100.2	208
10	105.2	181	20	104.7	178

**Table 5 sensors-24-00868-t005:** Uniform test arrangement for mixing level.

Number of Test	V (m/s)	N (r/min)	Crank Length (mm)	Shovel Knife Length (mm)	Shovel Knife Shape
1	0.1	115	175	230	straight edge straight shovel
2	0.15	165	165	200	pointed straight shovel
3	0.2	90	155	200	straight edge bending shovel
4	0.25	115	185	185	pointed bending shovel
5	0.3	140	155	230	straight edge bending shovel
6	0.35	165	185	215	straight edge bending shovel
7	0.4	90	175	215	straight edge bending shovel
8	0.45	140	165	185	straight edge straight shovel

**Table 6 sensors-24-00868-t006:** Simulation results of the uniform design test.

No. of Tests	V (m/s)	N (r/min)	Crank Length (mm)	Shovel Knife Length (mm)	Shovel Knife Shape	T (N·m)	M (N·m)
1	0.15	90	175	230	straight edge straight shovel	261.85	278.82
2	0.1	140	165	200	pointed straight shovel	71.424	74.033
3	0.2	65	155	200	straight edge bending shovel	98.350	143.045
4	0.25	90	185	185	pointed bending shovel	141.352	129.64
5	0.3	115	155	230	pointed bending shovel	100.536	147.21
6	0.35	140	185	215	straight edge bending shovel	246.332	247.03
7	0.45	65	175	215	pointed straight shovel	118.7567	132.73
8	0.4	115	165	185	straight edge straight shovel	110.0975	132.68

**Table 7 sensors-24-00868-t007:** Result of the torque–variance processing.

Impact Factor	Sum of Squares	Freedom	Equalized Square Sum	F	*p*
A	9821.7743	3	3273.9248	31,335.6287	0.0042
X1 × X4 (V × CD)	534.9116	1	534.9116	5119.7851	0.0089
X3 × X4 (QB × CD)	17,439.4995	1	17,439.4995	166,918.2159	0.0016
X4 × X4 (CD × CD)	1884.2737	1	1884.2737	18,034.8987	0.0047
residual	0.1045	1	0.1045		
Total variation	35,394.3611	7			

**Table 8 sensors-24-00868-t008:** Results of the torque regression analysis.

Related Parameters	Coefficient Size	Standard Error	t	*p*
Constant term	−378.8148			
A1	−54.5744	0.3247	−168.0905	0.0038
A2	−89.3959	0.3240	−275.9179	0.0023
A3	−13.95364	0.3238	−43.0905	0.0148
X1 × X4 (V × CD)	−0.350879	1.1279	−71.5527	0.0089
X3 × X4 (QB × CD)	0.02060835	2.1463	408.5563	0.0016
X4 × X4 (CD × CD)	−0.00333891	1.3152	−134.2941	0.0047

**Table 9 sensors-24-00868-t009:** Results of the bending moment–variance processing.

Impact Factor	Sum of Squares	Freedom	Equalized Square Sum	F	*p*
A	13,192.3274	3	4397.4425	2974.9768	0.0135
X3 (QB)	5.2143	1	5.2143	3.5276	0.3115
X1 × X2 (V × N)	194.6413	1	194.6413	131.6796	0.0553
X3 × X4 (QB × CD)	12,310.4262	1	12,310.4262	8328.3028	0.0070
residual	1.4781	1	1.4781		
Total variation	31,941.0968	7			

**Table 10 sensors-24-00868-t010:** Results of the bending moment regression analysis.

Related Parameters	Coefficient Size	Standard Error	t	*p*
Constant term	−256.7248			
A1	−61.5872	1.2185	−50.5441	0.0126
A2	−104.4889	1.2466	−83.8164	0.0076
A3	−10.2291	1.2165	−8.4085	0.0754
X3 (QB)	−0.0908	8.9411	−1.8782	0.3115
X1 × X2 (V × N)	−0.3886	4.7411	−11.4752	0.0553
X3 × X4 (QB × CD)	0.0138	6.4491	91.2595	0.0070

## Data Availability

All datasets used in this study are included in the manuscript.

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
