# Peer review of "The Optimized Design of Soil-Touching Parts of a Greenhouse Humanoid Weeding Shovel Based on Strain Sensing and DEM-ADAMS Coupling Simulation"

_sensors, 2024, doi:10.3390/s24030868_

Round 1

Reviewer 1 Report

Comments and Suggestions for Authors

1In the introduction, in view of the current tillage machinery in the grass entanglement, clay and other insurmountable problems, domestic and foreign scholars have carried out relevant research, such as the relevant research done by a certain scholar, it is recommended to illustrate the advantages and disadvantages of the structure in the following drawing.

22.1 crank connecting rod shovel mechanism design, the assembly of the key parts of the bulldozer shown in Figure 4 should be marked with a serial number, showing the detailed name of each part, and the installation method of the bearing is incorrect, and how to realize the axial fixation of the bearing is not explained.

3Fig. 5 The sketch of the crank connecting rod mechanism is not clear, which affects the viewing.

4The trajectory of the crank in Figure 6 is not clear, and the direction of the trajectory cannot be discerned.

52.2.3 In the EDEM-ADAMS coupling simulation, the amount of simulation data is small, although it is a preliminary judgment, but it cannot fully support the conclusion, whether to consider adding several more sets of data.

6In 2.3.1, "the travel wheel and sprocket are mounted on the bottom shaft and are supported by a bearing housing. "No Bearing housings are found in Figure 19.

7If the expression in Figure 22 is not clear, the circled position in the figure should be enlarged and displayed in a more conspicuous font.

8During the calibration in Figure 24, the oscilloscope is not clearly displayed.

9There are fewer references.

Comments on the Quality of English Language

no

Author Response

File Name: Review Report of Research Article  Sensors

Tittle:     Optimized Design of Soil Touching Parts of Greenhouse humanoid Weeding Shovel Based on Strain Sensing and DEM-ADAMS Coupling Simulation

Subject:        Authors response to the Editor

Respected Editor,

                        We would like to thank the Editor and Referee for reviewing and providing many useful suggestions to make our manuscript more effective. We revised the manuscript and tried our best by incorporating the suggestions of reviewers properly. In this response, we are presenting a point-by-point summary of the reviewer's Points.

Responses to reviewer 1.

Major concerns:

  1. In the introduction, in view of the current tillage machinery in the grass entanglement, clay and other insurmountable problems, domestic and foreign scholars have carried out relevant research, such as the relevant research done by a certain scholar, it is recommended to illustrate the advantages and disadvantages of the structure in the following drawing.

Response: Thank you for your valuable Point. We have revised the introduction and suggested advantages and disadvantages for its institutions

  1. 2.1 crank connecting rod shovel mechanism design, the assembly of the key parts of the bulldozer shown in Figure 4 should be marked with a serial number, showing the detailed name of each part, and the installation method of the bearing is incorrect, and how to realize the axial fixation of the bearing is not explained.

Response: We have labelled the assembly should be marked with the serial number and shown the detailed name of each part, also explaining how the bearing is axially secured.

  1. Fig. 5 The sketch of the crank connecting rod mechanism is not clear, which affects the viewing.

Response: We have made changes to Fig.5.

       4、The trajectory of the crank in Figure 6 is not clear, and the direction of the trajectory cannot be discerned.

Response: We have modified Fig. 6 and added the motion trajectory.

       5、2.2.3 In the EDEM-ADAMS coupling simulation, the amount of simulation data is small, although it is a preliminary judgment, but it cannot fully support the conclusion, whether to consider adding several more sets of data.

       Response: The simulation model in 2.2.3 only proposes a small number of groups for comparison, to facilitate the following field verification of the simulation, to verify the accuracy of the simulation after this paper has designed a uniform test to optimise the mechanism, the uniform test has a variety of data conclusions.

         6、In 2.3.1, "the travel wheel and sprocket are mounted on the bottom shaft and are supported by a bearing housing. "No Bearing housings are found in Figure 19.

         Response: We have changed the description, the walking wheel and the sprocket are installed on the bottom shaft and supported by the bearing seat. A side view has been added to Figure 19 with the location of the bearing seat marked out.

       7、If the expression in Figure 22 is not clear, the circled position in the figure should be enlarged and displayed in a more conspicuous font.

       Response: We have reworked Figure 22.

        8、During the calibration in Figure 24, the oscilloscope is not clearly displayed.

       Response: We have added an oscilloscope display.

      9There are fewer references.

       Response: We have added relevant references.

Reviewer 2 Report

Comments and Suggestions for Authors

1.Line 11-12, Suggest fixing the formatting at the end of the first line and the absence of a period in the sentence.

2.At the end of the introduction, the description of the research presented in this paper should be more detailed.

3.Line 92-95 and 97, There are formatting problems with the pictures and annotations.

4.Line 187 and 198, There are errors in the derivation formula and the variables lack specific explanations.

5.Line 465-466, Suggest changing "According to the field test results, the torque value of the tool at an operating speed of 0.2 m/s and 90 r/min, is 101.1829 N·m. " to "According to the field test results, the torque value of the tool at an operating speed of 0.2 m/s and 90 r/min is 101.1829 N·m. "

6.Line 569, The format of the significant horizontal critical value "F" is inconsistent with the content.

7.Line 594, Suggest changing "This paper reports the design of a human-like field shovel" to "This paper presents the design of a human-like field shovel".

8.The article lacks the description of the work effect. The description of soil condition parameters needs to be more detailed. It is necessary to analyze the impact of different soil conditions on the research results and compare them with other similar research results.

Comments on the Quality of English Language

English proficiency needs to be improved.

Author Response

File Name: Review Report of Research Article  Sensors

Tittle:     Optimized Design of Soil Touching Parts of Greenhouse humanoid Weeding Shovel Based on Strain Sensing and DEM-ADAMS Coupling Simulation

Subject:        Authors response to the Editor

Respected Editor,

                        We would like to thank the Editor and Referee for reviewing and providing many useful suggestions to make our manuscript more effective. We revised the manuscript and tried our best by incorporating the suggestions of reviewers properly. In this response, we are presenting a point-by-point summary of the reviewer's Points.

Responses to reviewer 2.

Major concerns:

  1. Line 11-12, Suggest fixing the formatting at the end of the first line and the absence of a period in the sentence.

Response: Thank you for your valuable Point. We have added a full stop in the appropriate place.

  1. At the end of the introduction, the description of the research presented in this paper should be more detailed.

Response: At the end of the introduction, We add the research done in this paper and describe it in detail.

  1. Line 92-95 and 97, There are formatting problems with the pictures and annotations.

Response: We have changed the formatting of the images in Lines 92-95 and 97.

4.Line 187 and 198, There are errors in the derivation formula and the variables lack specific explanations.

Response: I have explained the variables in the formulae in Lines 187 and 198, in the revised version of Line 178.

5.Line 465-466, Suggest changing "According to the field test results, the torque value of the tool at an operating speed of 0.2 m/s and 90 r/min, is 101.1829 N·m. " to "According to the field test results, the torque value of the tool at an operating speed of 0.2 m/s and 90 r/min is 101.1829 N·m.

Response: Thank you for your advice, We have taken your advice and corrected the content.

6.Line 569, The format of the significant horizontal critical value "F" is inconsistent with the content.

Response :I have changed the F format.

7.Line 594, Suggest changing "This paper reports the design of a human-like field shovel" to "This paper presents the design of a human-like field shovel".

Response: Thank you for your advice, We have taken your advice and corrected the content.

8.The article lacks the description of the work effect. The description of soil condition parameters needs to be more detailed. It is necessary to analyze the impact of different soil conditions on the research results and compare them with other similar research results.

Response: We have added and described in detail the specifics of soil modelling and the setting of soil parameters, which are currently only studied for the soils of the test site and will be continued for different soil environments in future work.

Reviewer 3 Report

Comments and Suggestions for Authors

1. It is necessary to present at least two design diagrams of soil cultivation devices used in small forms of farming using small-scale mechanization equipment.

2. When justifying the shape of the working body for tillage and modeling the process of movement of a soil particle represented by a material point, it is also necessary to consider the physical and mechanical properties of the soil.

3. It is necessary to present a methodology for determining the quality indicators of soil cultivation with an experimental working body.

4. It is necessary to justify the factors when conducting experimental studies of the developed working body.

5. It would be advisable to represent the energy costs for soil cultivation by the developed working body.

Author Response

File Name: Review Report of Research Article  Sensors

Tittle:     Optimized Design of Soil Touching Parts of Greenhouse humanoid Weeding Shovel Based on Strain Sensing and DEM-ADAMS Coupling Simulation

Subject:        Authors response to the Editor

Respected Editor,

                        We would like to thank the Editor and Referee for reviewing and providing many useful suggestions to make our manuscript more effective. We revised the manuscript and tried our best by incorporating the suggestions of reviewers properly. In this response, we are presenting a point-by-point summary of the reviewer's Points.

Responses to reviewer 3.

Major concerns:

  1. It is necessary to present at least two design diagrams of soil cultivation devices used in small forms of farming using small-scale mechanization equipment.

Response: Thank you for your valuable Point. In this paper , We have presented the design of the key components in Fig. 4, the overall mechanism design in Fig. 19 and added a side view to the original. The specific diagrams are shown below.

  1. When justifying the shape of the working body for tillage and modeling the process of movement of a soil particle represented by a material point, it is also necessary to consider the physical and mechanical properties of the soil.

Response: We have added the specifics of soil modelling, which takes into account soil properties such as density, shear modulus, etc.

  1. It is necessary to present a methodology for determining the quality indicators of soil cultivation with an experimental working body.

Response: This paper is designed to reduce the torque and bending moment generated during ploughing, and the main measurement objectives are reflected in the torque and bending moment measurements, and at the end, the measurement of shovel depth and width during ploughing and operational stability are added.

  1. It is necessary to justify the factors when conducting experimental studies of the developed working body.

Response: In this paper, different factors are set up for comparison using simulation model and the effect of each factor on bending moment and torque is verified by field test and finally a uniform test is designed to optimise the mechanism.

  1. It would be advisable to represent the energy costs for soil cultivation by the developed working body.

Response: Thank you for your valuable Point. This proposal makes a lot of sense. By monitoring and recording energy consumption in real time, we can better understand the energy efficiency of land farming and identify opportunities for improvement. We will follow up with more improvements.
